

# Developing a tablet-based brain-computer interface and robotic prototype for upper limb rehabilitation

Kishor Lakshminarayanan[1], Vadivelan Ramu[1], Rakshit Shah[2], Md Samiul Haque Sunny[3], Deepa Madathil[4], Brahim Brahmi[5], Inga Wang[6], Raouf Fareh[7] and Mohammad Habibur Rahman[3]

[1] Department of Sensors and Biomedical Tech, School of Electronics Engineering, Vellore Institute of Technology University, Vellore, Tamil Nadu, India
[2] Department of Orthopaedic Surgery, University of Arizona, Tucson, AZ, United States of America
[3] Department of Mechanical Engineering, University of Wisconsin-Milwaukee, Milwaukee, WI, United States of America
[4] Jindal Institute of Behavioural Sciences, O.P. Jindal Global University, Haryana, India
[5] Electrical Engineering, Collège Ahuntsic, Montreal, QC, Canada
[6] Department of Occupational Science & Technology, University of Wisconsin-Milwaukee, Milwaukee, WI, United States of America
[7] Department of Electrical and Computer Engineering, University of Sharjah, Sharjah, United Arab Emirates

Corresponding author
Kishor Lakshminarayanan,
kishor.ln@vit.ac.in

## ABSTRACT

**Background**. The current study explores the integration of a motor imagery (MI)-based BCI system with robotic rehabilitation designed for upper limb function recovery in stroke patients.

**Methods**. We developed a tablet deployable BCI control of the virtual iTbot for ease of use. Twelve right-handed healthy adults participated in this study, which involved a novel BCI training approach incorporating tactile vibration stimulation during MI tasks. The experiment utilized EEG signals captured *via* a gel-free cap, processed through various stages including signal verification, training, and testing. The training involved MI tasks with concurrent vibrotactile stimulation, utilizing common spatial pattern (CSP) training and linear discriminant analysis (LDA) for signal classification. The testing stage introduced a real-time feedback system and a virtual game environment where participants controlled a virtual iTbot robot.

**Results**. Results showed varying accuracies in motor intention detection across participants, with an average true positive rate of 63.33% in classifying MI signals.

**Discussion**. The study highlights the potential of MI-based BCI in robotic rehabilitation, particularly in terms of engagement and personalization. The findings underscore the feasibility of BCI technology in rehabilitation and its potential use for stroke survivors with upper limb dysfunctions.

# INTRODUCTION

Globally, stroke affects over 15 million people annually (*World Stroke Organization, 2022*), with upper limb impairments being a common consequence. The burden of upper limb

dysfunctions in stroke survivors presents a significant challenge in healthcare, both socially and economically. These dysfunctions not only limit individual independence and quality of life but also impose a massive socio-economic burden, with costs in the United States alone exceeding $100 billion per year (*Girotra et al., 2020*). As the population ages, there is an urgent need for improved rehabilitation care for individuals with upper limb dysfunctions. Currently, physical therapy is the predominant form of treatment for stroke patients, typically involving human therapists who aid in the recovery of the body parts affected by the stroke. The advent of robotic rehabilitation has enhanced these traditional physical therapy methods, offering innovative exercises that human therapists alone cannot provide (*Laut, Porfiri & Raghavan, 2016*). Research has demonstrated that robotic rehabilitation significantly aids in the improvement of hemiparetic upper extremity impairments in individuals who have experienced a chronic stroke (*Bertani et al., 2017*).

In the context of robot-based rehabilitation, brain-computer interface (BCI) technology offers a groundbreaking approach for stroke patients, especially those with severe upper limb dysfunctions (*Ang et al., 2009*). BCI system based on non-invasive electroencephalography (EEG) functions by detecting and interpreting the patient's brain signals associated with motor imagery (MI) (*Lotte, 2009*; *Wolpaw, Millán & Ramsey, 2020*; *Birbaumer, 2006*). MI involves the mental simulation of specific motor sequences without actual muscle movement (*Andrade et al., 2017*). MI activates several cortical areas akin to those involved in motor execution, including the primary motor cortex, premotor cortex, and somatosensory association cortices (*Lorey et al., 2014*; *Ehrsson, Geyer & Naito, 2003*; *Sauvage et al., 2013*). Specifically, MI elicits an event-related desynchronization (ERD) in the alpha band (8–12 Hz) and the beta band (13–30 Hz) during the mental visualization of motor movements (*Pfurtscheller & Neuper, 1997*; *Jeon et al., 2011*; *McFarland et al., 2000*). MI-based BCI training is extensively utilized by individuals aiming to enhance their motor skills, including stroke survivors (*Ang et al., 2009*). When a patient imagines moving their impaired limb, the BCI system identifies these specific motor imagery signals which are crucial as it reflects the patient's intent to move, despite the physical inability to do so due to stroke-induced impairments.

The innovative aspect of BCI in robotic rehabilitation lies in its integration with robotic systems. For instance, the iTbot is an end-effector-type robot specifically designed for the rehabilitation and reinforcement of motor skills in individuals suffering from neuromuscular ailments, such as those resulting from stroke (*Khan et al., 2022*). The iTbot assists in the rehabilitation training of elbow flexion and extension motions. If integrated with BCI, the system will capture the patient's motor intention, and translate these signals into commands. The BCI commands will then be communicated to the iTbot, which in turn will assist the patient in performing the intended movement. Such a symbiotic relationship between the patient's brain signals and the robot's mechanical assistance will foster an interactive environment for rehabilitation, enabling the patient to partially regain motor control and continue their rehabilitation exercises, facilitating improved recovery and functional independence.

For training the BCI system, combining tactile stimulation with MI has been found to enhance MI effectiveness. Research conducted by *Mikula et al. (2018)* revealed that during

a reaching task, hand proprioception was improved with the addition of vibrotactile stimulation. Electrophysiological studies also support that MI paired with sensory stimulation significantly increased motor evoked potentials (*Mizuguchi et al., 2015*). Additionally, *Ramu & Lakshminarayanan (2023)* demonstrated that a brief vibratory stimulation on the digit before MI enhanced ERD response compared to without vibration and increased digit discrimination when classified using a machine learning algorithm.

Therefore, the purpose of the current study is to integrate motor imagery and a rehabilitation robot to leverage the motor intent captured by the MI-BCI system to control the iTBot. To achieve this, we aim to develop a prototype of a BCI system where the training will be performed with vibration stimulation. Furthermore, the BCI system will be used to control a virtual iTbot game for smartphone/tablet in real-time.

## METHODS

### BCI system development

The BCI system was developed using EEG signals acquired from the OpenBCI Cyton Daisy module (OpenBCI, Inc., Brooklyn, NY, USA), featuring a gel-free 16-electrode EEG cap. The system adheres to the international 10-20 system for electrode placement, with electrodes located at FP1, FP2, C3, C4, CZ, P3, P4, PZ, O1, O2, F7, F8, F3, F4, T3, and T4. In this study, the gel-free cap technology was employed, eliminating the need for conductive gel application while ensuring each electrode site maintained a good impedance level for high-quality EEG signal acquisition. Impedance level was checked in the OpenBCI GUI. The ground and reference electrodes were optimally positioned to enhance signal clarity and minimize artifacts. The cap was securely placed on each subject's scalp, ensuring consistent electrode contact and stability throughout the data collection. A short sensory stimulation was applied to the left or right wrist during the BCI training stage. The sensory stimulation was applied *via* vibration through a flat vibration micro motor (Sunrobotics, Gujarat, India). The vibration motor produced a white-noise vibration, with its frequency spectrum filtered to range from 0 to 500 Hz (Fig. 1).

A video game was developed to establish a virtual environment that mimics movements generated by brain signals through both visual and auditory feedback. Created using the Unity game engine (Unity Technologies, San Francisco, CA, USA), the game enabled subjects to control a virtual iTbot robot. The iTbot is a three degrees-of-freedom end-effector type robot used for upper limb rehabilitation. A digital version of the iTbot was built inside Unity with the same degrees-of-freedom. The game was built with Android studio in Unity as a smartphone/tablet application (Fig. 2). The game was run on a tablet that the subjects held in their hands and controlled using BCI. The goal within the game was to pop balloons positioned around the robot using its end effector.

### Subjects

The study involved twelve right-handed healthy adults, including five females and seven males, aged between 21 and 39 years. Each participant confirmed verbally that they had no history of upper limb injuries, musculoskeletal disorders, or neurological conditions. None of the subjects had previous experience with motor imagery.

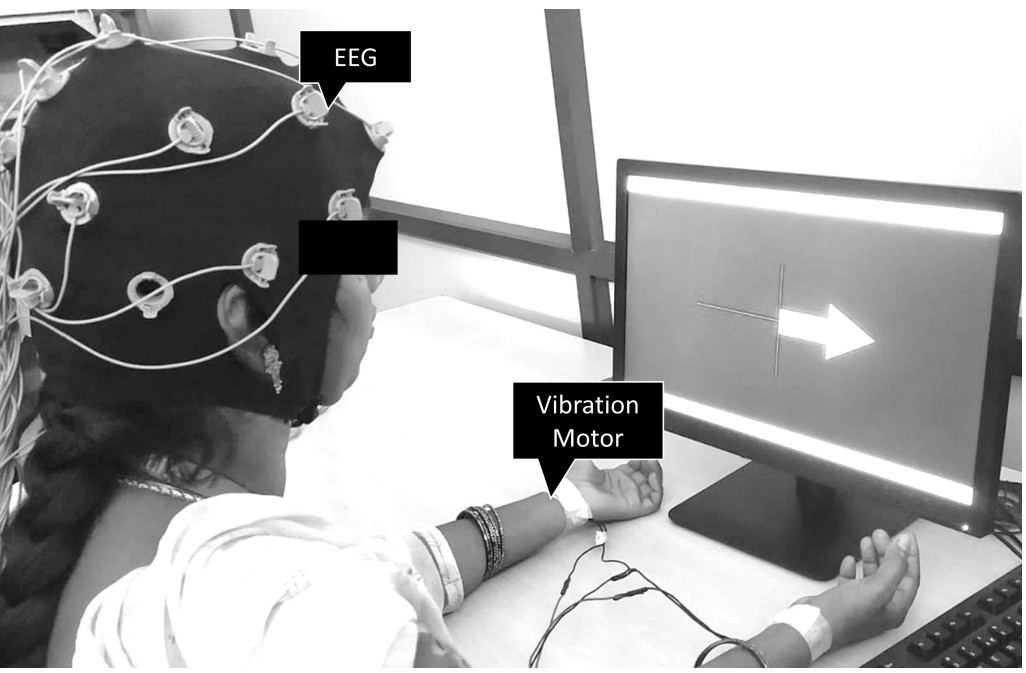

**Figure 1** Subject wearing the OpenBCI gel-free cap and performing BCI training with vibration motors mounted on left and right wrists.

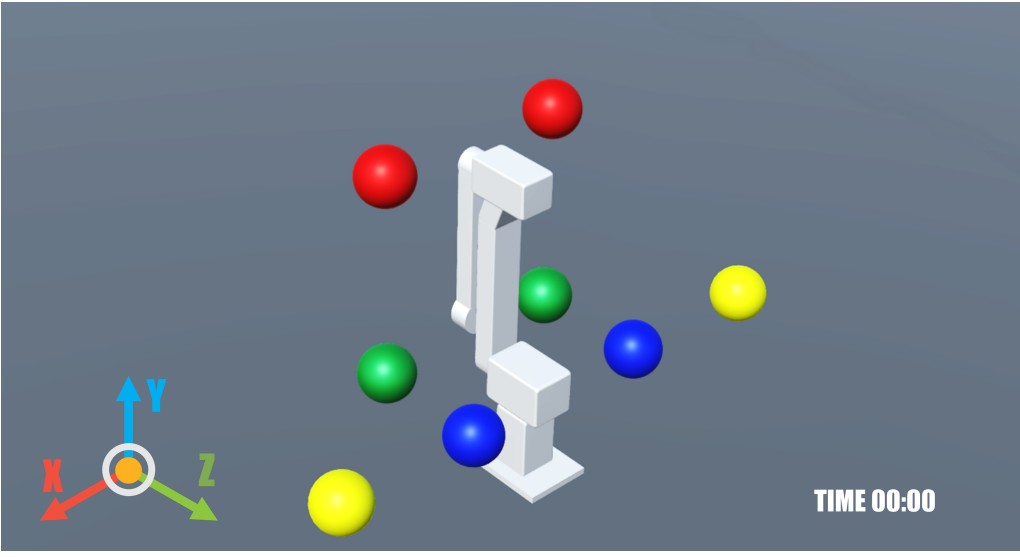

**Figure 2** Game screen as seen on the tablet showing a virtual iTbot.

The Vellore Institute of Technology Review Board approved the study's protocol (VIT/IECH/IX/Mar03/2020/016B). Before participating in the experiment, all subjects read and signed a written informed consent form.

## Experimental procedure

The study adopted the motor imagery CSP scenarios from OpenVibe (Inria Hybrid Team, Campus de Beaulieu, France) and made modifications to it. There were several stages to the study namely the signal verification stage where the signal quality was checked, followed by the training stage where the calibration signals were obtained while subjects were asked to imagine moving their left or right hand based on a cue with a short sensory stimulation applied to the corresponding left or right wrist when the cue was displayed. Following the training stage was the testing stage where subjects had to imagine moving their left and right hand based on the cue with feedback in the form of a bar graph that moved left or right. Finally, the subjects played a game with real-time control of a virtual iTbot using motor imagery.

## Signal verification stage

The OpenVibe Acquisition Server was used to connect to the OpenBCI EEG. Once the device was connected the signal verification was accomplished in the mi-csp-0-signal-monitoring scenario in OpenVibe. The scenario had a band pass filter between 1 to 40 Hz and filter order 4 that was applied on the raw signals. Running the scenario displayed a live plot with both the raw and filtered signals from each channel, where they could be checked for noise and artifacts. Once the signals were verified the scenario was stopped.

## Training stage
### Signal acquisition

Following the verification stage, training the BCI system was performed by running three scenarios consecutively in OpenVibe. In the mi-csp-1-acquisition scenario, EEG signals were acquired to train the classifier to discriminate between the left and right hand imagined movements. The scenario was configured to have 50 trials each for the two classes namely, left and right hand, randomly displayed to each subject. An initial baseline period of 20 s was followed by the trials with a 30 s break between every 10 trials. Each trial was 5 s long with the cue in the form of arrows pointing left or right to indicate the hand to be imagined displayed to the subject for the first 1.25 s followed by which the subjects were instructed to imagine moving the corresponding hand. EEG signals were recorded at 250 Hz continuously during the trials. Following the randomized trials, a confusion matrix showing the performance of the classifier was generated in the mi-csp-5-replay scenario.

The short sensory stimulation was applied to the corresponding left or right wrist while the cue was displayed for the duration of the cue. The vibrators were connected to an ESP8266 module that was connected to the OpenVibe scenario using the Open Sound Control (OSC) communication protocol.

### Common spatial pattern training

Following the signal acquisition, the mi-csp-2-train-csp scenario was run to train common spatial pattern (CSP) to produce a spatial filter that maximizes the difference between the signal of the two classes. CSP is a mathematical approach designed to decompose a multivariate signal into distinct subcomponents. The essence of this procedure is to identify and separate components that exhibit the greatest variance differences when comparing

two specific time windows. CSP calculated $w^T$ (Eq. (1)), which is crucial for maximizing the variance radius between the two selected time windows (*Antony et al., 2022*).

$$w = \mathrm{argmax}_w \|wX_1\|^2 / \|wX_2\|^2. \tag{1}$$

The EEG signal obtained in the previous scenario was filtered in a large frequency band (8–30 Hz) to cover both alpha and beta frequency band range. The filtered continuous EEG data was then segmented to 4-second long epochs. The epochs from the left hand and right hand trials were segregated and the CSP was computed. CSP is vital for extracting signal components that are specifically useful for binary classification configurations. The significance of CSP lies in its ability to identify components of the signal whose variance is most informative for the classification task at hand. Studies (*Zhang et al., 2020*; *Singh, Gautam & Sharan, 2022*; *Yu et al., 2019*) have shown that these components often possess superior spectral characteristics compared to the raw channels. Consequently, employing the CSP node can lead to enhanced classification accuracy, making it a critical element in the analysis of time series data for binary classification purposes.

### Linear discriminant analysis

The motor intention of each subject was classified in the mi-csp-3-classifier-trainer scenario. In our study, we evaluated the three classifiers available in the OpenVibe toolbox, specifically the linear discriminant analysis (LDA), support vector machine (SVM), and multilayer perceptron (MLP). After conducting comparative analyses, the results demonstrated that the LDA classifier showed the highest classification accuracy of 63.32% while MLP showed 59.86% and SVM showed 63.26%. Based on these findings, we selected the LDA classifier for our study. LDA, a method in supervised machine learning, excels in classifying input data into two or more distinct classes by linearly mapping features with categorical labels. Its primary function is to enhance class separation by maximizing the ratio of between-class variance to within-class variance. This approach not only improves the clarity of class distinctions but also maintains computational efficiency, making it ideal for real-time applications like BCI. LDA's statistical basis provides a clear interpretive framework, ensuring both high accuracy in classification tasks and straightforward interpretability, essential in fields like neuroscience and rehabilitation engineering.

The CSP spatial filter trained in the previous scenario is applied to the band pass filtered (8–30 Hz) EEG signals. Following which the signal was epoched to extract a four-second segment of the signal, commencing half a second after the display of the cue to the user. Post this step, the signal is divided into one-second blocks, occurring every 16 s. The logarithmic band power of these segments is then calculated using two 'Simple DSP' boxes along with the 'Signal Average' box. The process ultimately facilitates the conversion of the signal matrices into feature vectors. The extracted features were then used to train the LDA classifier with a 5-fold cross validation, where the data was divided into five parts, with four parts used for training and one part for testing, iteratively. At the end of the training the classifier performance was displayed as a percentage for both training and testing. The classifier trainer also produced a configuration file to be used for the online testing stage.

## Testing stage

Once the CSP and the LDA classifier was trained, the mi-csp-4-online scenario (Fig. 3) was run where the subjects underwent trials similar to the training stage with the cue based motor imagery trials but with real-time feedback in the form of a horizontal bar graph that shifted left or right based on the classified motor intent.

The game interfaced in real-time with the 'mi-csp-4-online' scenario *via* OSC, receiving live classification percentages for imagined movements of the left and right hands. The OSC control enabled a wireless link between the laptop running OpenVibe and the tablet running the game. In this interactive setting, participants could navigate the robotic arm along the $x$, $y$, and $z$ axes using the toolbar on the side of the game screen that displayed the X, Y, and Z buttons. After choosing an axis, the participants were able to direct the robot's movement through motor imagery (Fig. 4). This involved imagining the movement of either their left or right hand, which corresponded to the robot's movement direction. For the $x$ axis, this meant moving left or right; for the $y$ axis, it meant moving forward or backward; and for the $z$ axis, it involved moving up or down. The robotic arm responded to these imagined movements, initiating motion when the classification percentage for an imagined movement reached a 60% threshold, and continued moving as long as the percentage remained above this threshold. To add an element of challenge and engagement, the game included a timer, encouraging subjects to pop all the balloons as quickly as possible. This gaming task served as the concluding activity for the subjects.

## RESULTS

The current study focused on evaluating the effectiveness of a BCI system integrated with robotic rehabilitation technology for aiding stroke survivors with upper limb dysfunctions. We evaluated the classification accuracy of motor intention detection using the LDA based method during the motor imagery training session through a confusion matrix, which is crucial in understanding how accurately the BCI system can interpret motor imagery signals.

The comparison of training and testing accuracy during the motor imagery training session for each subject is visually represented in the bar graph (Fig. 5). We noticed that the testing accuracy was higher than the training accuracy for almost all participants, but the difference wasn't very large. The results could indicate that our model was generalizing well rather than just memorizing the training data. We used a 5-fold cross-validation method to help prevent overfitting.

The average confusion matrix across all subjects was calculated to provide insights into the classification performance during the motor imagery training session (Fig. 6). The provided metrics pertain to the performance of a classification model in distinguishing between left and right MI signals. A true positive (TP) rate of 0.6333 indicates that the model correctly identified approximately 63.33% of the MI signals, demonstrating its effectiveness in recognizing instances of MI. On the other hand, a false positive (FP) rate of 0.3658 reflects that around 36.58% of the time, left MI signals were incorrectly classified as right MI. The false negative (FN) rate of 0.3675 suggests that the model missed

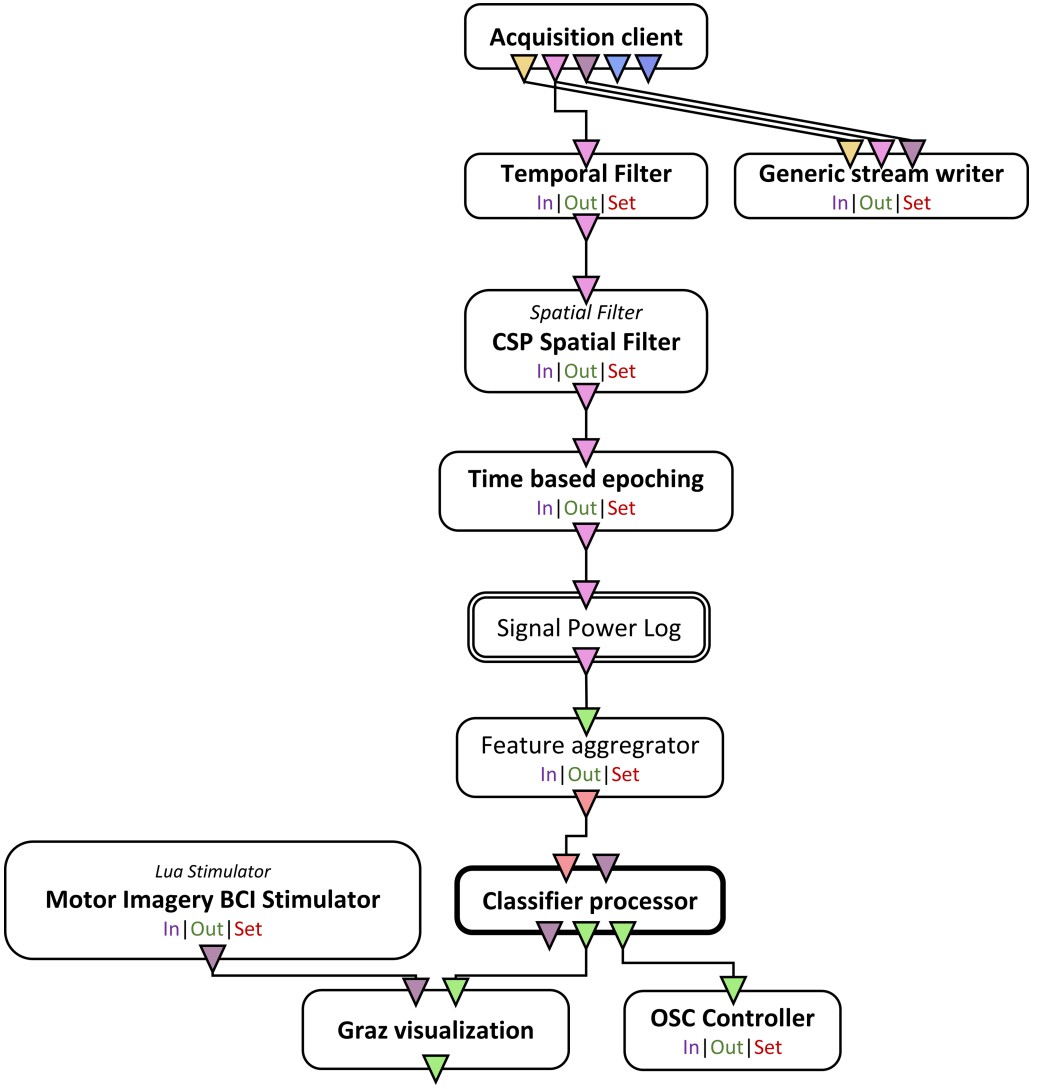

**Figure 3** Schematic of the online scenario used once the CSP spatial filter and the classifier is trained.

about 36.75% of actual left MI signals, failing to detect them correctly. Finally, the true negative (TN) rate of 0.6325 shows that the model correctly identified right MI signals approximately 63.25% of the time.

The accuracy of MI signal detection directly impacts the effectiveness of robotic systems like the iTbot in assisting patients with motor skill rehabilitation. While the results show promise, they also indicate areas for improvement, particularly in reducing false positives and negatives, which are vital for ensuring precise and beneficial rehabilitation exercises.

## DISCUSSIONS

In the current study, healthy participants were engaged in a task that involved imagining the movement of their upper limbs. This motor imagery was used to control a virtual

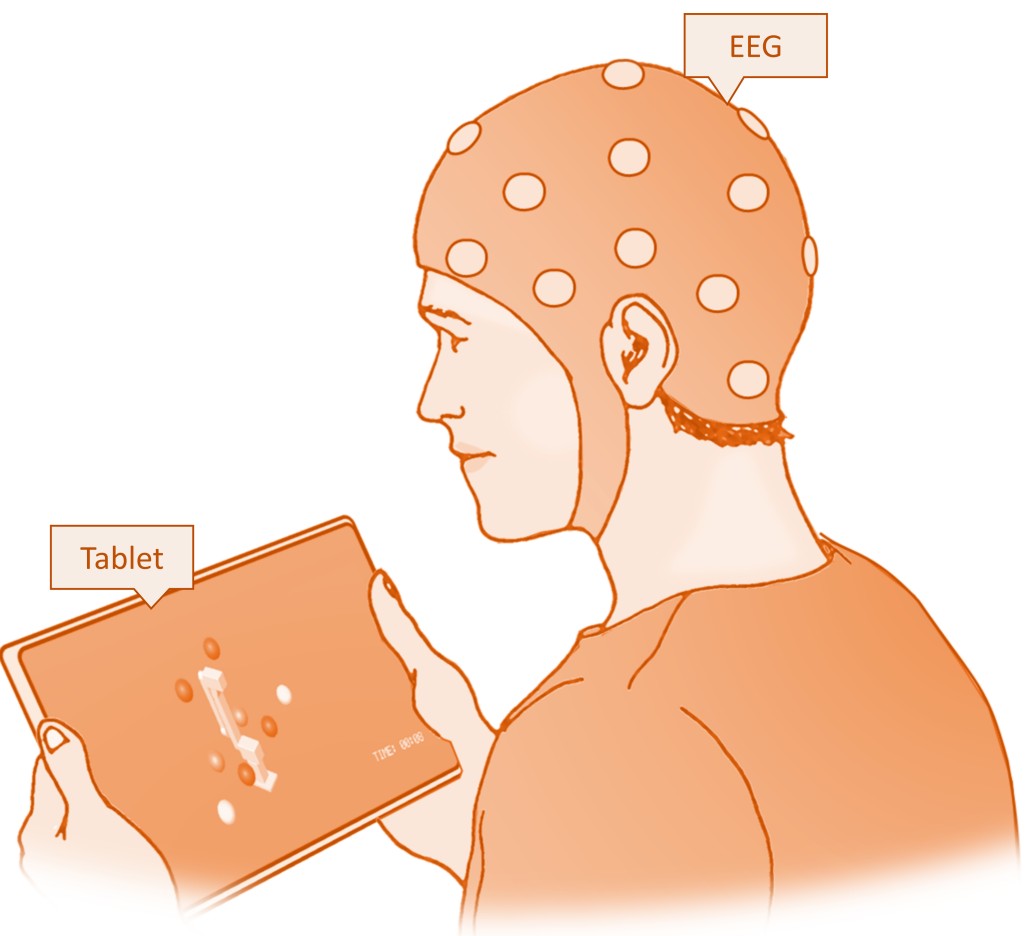

**Figure 4** Experimental setup for the testing session showing the subject playing the game on a tablet.

rehabilitation robot's movement along the three axes in a simulation. The aim was to explore the potential of such a system in supporting rehabilitation of upper limb functions. EEG signals were captured using a cap equipped with OpenBCI gel-free electrodes. These signals were processed to produce control commands, which in turn manipulated the movements of the virtual robot in the simulation. The study not only focused on the efficiency of signal processing and the accuracy of movement classification but also on how sustainably participants could control the application in real-time.

The study integrated short vibrotactile stimulation just prior to MI during the training phase. This strategic inclusion was guided by the encouraging outcomes observed in our preceding research (*Ramu & Lakshminarayanan, 2023*). This earlier study revealed that vibrotactile stimulation could amplify event-related desynchronization in the beta-band within the contralateral sensorimotor area during MI, which reflects enhancement in motor tasks (*Khanna & Carmena, 2015*). Our past findings indicated that such vibrotactile stimulation might exert its effects on the sensorimotor cortex, possibly by enhancing proprioception (*Rizzolatti, Luppino & Matelli, 1998*; *Mikula et al., 2018*). Proprioception,

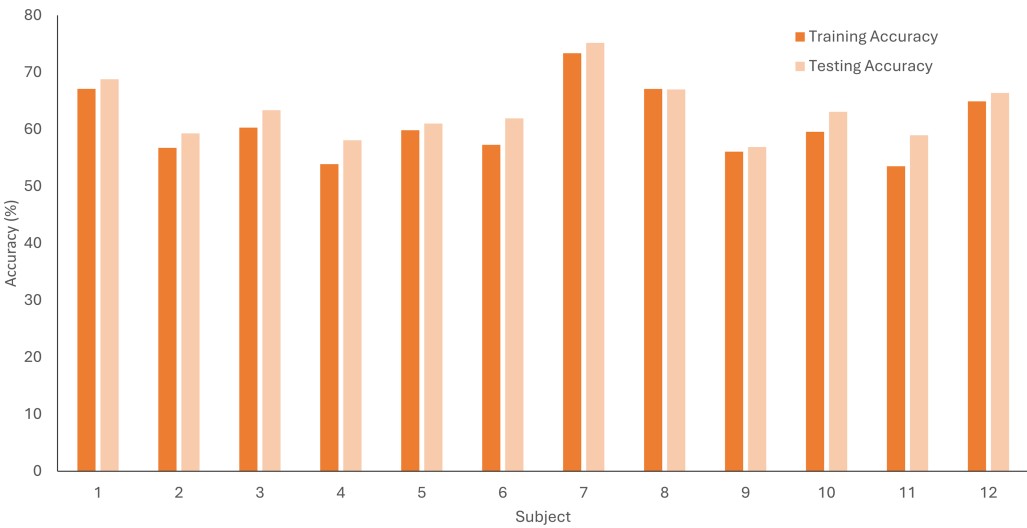

**Figure 5** LDA accuracy percentage for training and testing data sets averaged across 5-folds for each subject for the motor imagery training data.

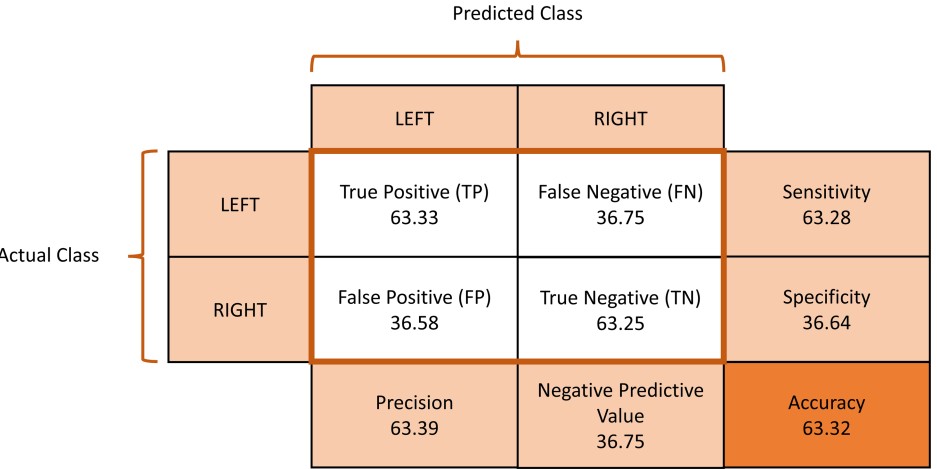

**Figure 6** Average confusion matrix of the motor imagery training session classification performance.

the awareness of the position and movement of one's body parts, has been documented to boost corticospinal excitability during MI (*Vargas et al., 2004*). This aspect is critical, considering proprioception and tactile sensation share neural pathways, particularly in the posterior parietal neurons responsible for high-level spatial representations (*Rizzolatti, Luppino & Matelli, 1998*). Moreover, our earlier investigations underscored the role of the brain in utilizing diverse sensory information to construct an internal representation of the environment (*Knill & Pouget, 2004*; *Ernst & Bülthoff, 2004*). This internal representation is essential for motor imagery, relying on it to anticipate future sensory and motor states during the imagination of movement (*Gentili & Papaxanthis, 2015*; *Nicholson, Keogh &*

*Choy, 2018*). The vibrotactile stimulation in our study was anticipated to provide essential sensory input, thereby enhancing this internal representation through proprioception. Specifically, the stimulation of the digits aimed to enrich the ERD response during MI by bolstering the sensory information needed to generate this internal representation (*Rulleau et al., 2018*). Therefore, the application of short vibrotactile stimulation before MI in our online BCI training phase was a decision rooted in these empirical insights. We hypothesized that this approach would improve the efficiency of the BCI by enhancing the clarity and precision of the MI-related neural signals, as suggested by our previous study's results in offline BCI classification.

Incorporating control mechanisms in therapeutic devices is pivotal for the efficacy of robotic-assisted rehabilitation training. Specifically, patients suffering from acute hemiplegia show promising improvements when engaged in a passive training regimen. This approach involves the automated guidance of the impaired limb following a set trajectory, enabling passive yet repetitive movements (*Rahman et al., 2012*). The design of these rehabilitation systems is inherently more complex than conventional robotic manipulators, given the intricate non-linear dynamics of the robots, the presence of unforeseen external disturbances, and the unique viscoelastic characteristics of human joints (*Wu, Chen & Wu, 2019*). To refine the precision of position control in repetitive reaching exercises, various innovative control strategies have been implemented in rehabilitation robots. These strategies range from adaptive control (*Feng et al., 2016*) and flatness-based control (*Brahmi et al., 2021*) to EMG-based control (*Rahman, Ochoa-Luna & Saad, 2015*), admittance control (*Ayas & Altas, 2017*), and a blend of fuzzy logic and backstepping control (*Chen, Li & Chen, 2016*). Building on these developments, evaluating the feasibility of integrating BCI control with an end-effector type robot presents a novel avenue in robotic rehabilitation. The addition of BCI control could offer a more personalized and responsive rehabilitation experience. By interpreting the patient's neural signals directly, the BCI system can potentially provide real-time adjustments to the robot's movements, aligning more closely with the patient's intended motor functions. This synergy between neural intent and robotic movement could not only enhance the effectiveness of the therapy but also potentially accelerate the patient's recovery by promoting active participation and neuroplasticity. However, this integration poses significant challenges, including the need for robust signal processing algorithms to accurately interpret neural signals and the development of adaptive control strategies that can seamlessly respond to these signals in a dynamic rehabilitation environment.

In our study, we developed a virtual version of an end-effector robot designed for a unique game where participants were tasked with popping balloons using the robot. The selection of classifiers and signal processing techniques for our virtual prototype drew upon extensive research, guided by insights from previously published systematic reviews (*Palumbo et al., 2021*; *Prashant, Joshi & Gandhi, 2015*; *Camargo-Vargas, Callejas-Cuervo & Mazzoleni, 2021*). However, the final design of our model was also shaped significantly by observations made during the training sessions with participants. It is crucial to underscore that our prototype serves as an initial proof of concept. Its primary purpose is to assess the viability and potential enhancements in the design of BCIs for upper limb rehabilitation.

Interestingly, we observed that participants displayed greater engagement and motivation during the video game interaction phase compared to the initial setup and training phases. Many studies have concentrated on the need to give sufficient feedback to individuals during training in the MI and BCI loops (*Ono, Kimura & Ushiba, 2013*; *Vuckovic & Osuagwu, 2013*). The feedback appears to be connected to the development of the MI ability since it improves the command classification accuracy in the BCI (*González-Franco et al., 2011*). Feedback further enhances the training regimen, makes it more interactive, and helps improve the subject's interest in and participation in the activity. The most commonly used and widely investigated method of feedback is visual, which requires the participant to focus attention and concentrate on the BCI system (*López-Larraz et al., 2011*). Our findings reinforce the idea that interactive feedback is not only engaging but also facilitates a more effective training process and a stronger connection between the BCI system and the user.

Our study faced several limitations, one of which is the classification accuracy percentage, though above the statistical chance level, is still not high as seen in other similar MI studies. This could be due to the limited number of participant and the inter-subject variability in EEG signals. Furthermore, the study only tested healthy participants making it hard to generalize the BCI results. However, the aim of the study is to develop a prototype for a virtual iTBot rehabilitation robot controlled *via* BCI which we were able to accomplish. Future studies will focus on improving the BCI classification accuracy by employing other algorithms and testing patient population as well.

## CONCLUSION

Our study explored the integration of BCI technology with a virtual end-effector robot in a game setting. The results of our study highlight the potential of BCI technology in offering a more engaging, personalized rehabilitation experience. Participants showed increased motivation during the gaming phase, underscoring the importance of interactive feedback during rehabilitation exercises. While it opens promising avenues for future research in integrating rehabilitation robots with BCI control, it also highlights the challenges in achieving higher classification accuracy.

### Funding
The authors received no funding for this work.

### Competing Interests
The authors declare there are no competing interests.

### Author Contributions
- Kishor Lakshminarayanan conceived and designed the experiments, performed the experiments, analyzed the data, performed the computation work, prepared figures and/or tables, authored or reviewed drafts of the article, and approved the final draft.

- Vadivelan Ramu conceived and designed the experiments, performed the experiments, analyzed the data, performed the computation work, prepared figures and/or tables, and approved the final draft.
- Rakshit Shah conceived and designed the experiments, performed the experiments, analyzed the data, prepared figures and/or tables, and approved the final draft.
- Md Samiul Haque Sunny conceived and designed the experiments, performed the experiments, analyzed the data, performed the computation work, prepared figures and/or tables, and approved the final draft.
- Deepa Madathil performed the experiments, analyzed the data, authored or reviewed drafts of the article, and approved the final draft.
- Brahim Brahmi performed the experiments, authored or reviewed drafts of the article, and approved the final draft.
- Inga Wang performed the experiments, analyzed the data, prepared figures and/or tables, authored or reviewed drafts of the article, and approved the final draft.
- Raouf Fareh performed the experiments, authored or reviewed drafts of the article, and approved the final draft.
- Mohammad Habibur Rahman conceived and designed the experiments, performed the experiments, analyzed the data, prepared figures and/or tables, authored or reviewed drafts of the article, and approved the final draft.

### Ethics

The following information was supplied relating to ethical approvals (*i.e.*, approving body and any reference numbers):

The Vellore Institute of Technology Review Board approved the study's protocol (VIT/IECH/IX/Mar03/2020/016B). Before participating in the experiment, all subjects read and signed a written informed consent form.

### Data Availability

The raw data is available in the Supplemental File.

### Supplemental Information

Supplemental information for this article can be found online at http://dx.doi.org/10.7717/peerj-cs.2174#supplemental-information.

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
