# Peer review of "Developing a tablet-based brain-computer interface and robotic prototype for upper limb rehabilitation"

_PeerJ Computer Science, doi:10.7717/peerj-cs.2174_

## Round 0.1 · original submission · Minor Revisions

The article is well structured but need some improvements. Also please clarify the accuracy as it is not too high!

Reviewer 1 ·

Basic reporting

The main idea of the entire manuscript is interesting. If I understood it correctly, the authors developed a tablet deployable BCI control system incorporating tactile vibration simulation on MI tasks. The collected EEG signals were subsequently trained using LDA algorithm for signal classification here is left of right motor movement. Literature review is well conducted, which shows sufficient background of this study.

Experimental design

I do have a few concerns on the overall study design. This study only has 12 healthy subjects. Not mentioning the size of the study is too small (leading to 2 - 3 subjects per CV fold), some unhealthy subjects at least should be recruited for control experiment. The EEG signals can be quite different between healthy and unhealthy patients. I do believe this is a key piece in any research design.

Validity of the findings

In terms of the results, what is the underlying data distribution (e.g., how many positive and negative classes)? Likely there will be some data imbalance, how is the performance of the chosen ML algorithm compared to baseline results, which can be something simple like majority vote. For figure 5, why reporting accuracies on each subject level, I think you should look at all subject from the same group as a whole. In addition, 63.33% TP rate does not seem to impressive. Why not exploring other types of models, for example, tree-based models to see if there're any non-linear relationship can be captured? I do think there're a good amount of work still needs to be put into this manuscript.

Reviewer 2 ·

Basic reporting

The writing is clear and easy to follow, with professional English used throughout. The literature references and field background/context are sufficient. The article structure is professional, but it would be beneficial to share raw data. The results are self-contained and relevant to the hypothesis.

Experimental design

1. The research question is well-defined and relevant to the journal's Aims and Scope, exploring the integration of BCI technology with a virtual end-effector robot in a game setting for upper limb dysfunctions. The research fills an identified knowledge gap in the field.
2. The investigation is original primary research, but the sample size of 12 subjects seems a bit small. Consider using open datasets as a secondary dataset to further validate the results.
3. The methods are described with sufficient detail, but it would be beneficial to include more information to replicate the experiment.

Validity of the findings

1. The data provided is robust, but the use of LDA for classification without exploring other baseline models may limit the validity of the findings.
2. The results are statistically sound, but the conclusions may be overstated given the moderate accuracy of around 55-70%.

Additional comments

1. The paper would benefit from a more comprehensive comparison of different machine learning models, including their computational efficiency and inference time.
2. The discussion of the results could be improved by addressing the unexpected phenomenon of higher testing accuracy than training accuracy for 11 subjects.
3. The paper's claims about the complexity of developing effective BCI-based therapeutic tools are not fully supported by the experimental results, and the accuracy achieved does not seem to justify the claim of "promising avenues for future research".

---

## Round 0.2 · accepted · Accept

The paper was well improved and can be accepted.

Reviewer 2 ·

Basic reporting

The manuscript is well-written in clear and professional English, with a professional structure, figures, and tables. The raw data has been shared as requested.

Experimental design

The research question is well-defined, relevant, and meaningful, and the investigation is original primary research within the journal's Aims and Scope. The methods are described with sufficient detail to replicate the experiment.

Validity of the findings

The underlying data appears robust, statistically sound, and controlled, and the conclusions are well-stated and linked to the original research question. The authors have addressed the limitations of the study, including the small sample size and moderate classification accuracy.

Additional comments

I am pleased to see that the authors have thoroughly addressed all the issues raised in my previous review. The manuscript is now clear, well-organized, and easy to follow. The authors have provided additional information to replicate the experiment, including the raw data. The limitations of the study are acknowledged and discussed, and the conclusions are more balanced and nuanced.